# An Integrated Statistical-Machine Learning Approach for Runoff Prediction

Abhinav Kumar Singh [1], Pankaj Kumar [1], Rawshan Ali [2], Nadhir Al-Ansari [3,*], Dinesh Kumar Vishwakarma [4,*], Kuldeep Singh Kushwaha [5], Kanhu Charan Panda [6], Atish Sagar [7], Ehsan Mirzania [8], Ahmed Elbeltagi [9], Alban Kuriqi [10,11] and Salim Heddam [12]

1   Department of Soil and Water Conservation Engineering, G. B. Pant University of Agriculture and Technology, Pantnagar 263145, India; avi4913@gmail.com (A.K.S.); pankaj591@gmail.com (P.K.)
2   Department of Petroleum, Koya Technical Institute, Erbil Polytechnic University, Erbil 44001, Iraq; rawshan.ali@epu.edu.iq
3   Department of Civil, Environmental and Natural Resources Engineering, Lulea University of Technology, 97187 Lulea, Sweden
4   Department of Irrigation and Drainage Engineering, G. B. Pant University of Agriculture and Technology, Pantnagar 263145, India
5   Centre for Water Engineering and Management, Central University of Jharkhand, Ranchi 835205, India; kuldeepkushwaha@hotmail.com
6   Department of Agricultural Engineering, Institute of Agricultural Sciences, Banaras Hindu University, Varanasi 221005, India; kanhucharan.bm@gmail.com
7   Division of Agricultural Engineering, ICAR—Indian Agriculture Research Institute, New Delhi 110012, India; atishmicky.sagar@gmail.com
8   Department of Water Engineering, Faculty of Agriculture, University of Tabriz, Tabriz 5166616471, Iran; e.mirzania99@ms.tabrizu.ac.ir
9   Agricultural Engineering Department, Faculty of Agriculture, Mansoura University, Mansoura 35516, Egypt; ahmedelbeltagy81@mans.edu.eg
10  CERIS, Instituto Superior Técnico, University of Lisbon, 1649-004 Lisbon, Portugal; alban.kuriqi@tecnico.ulisboa.pt
11  Civil Engineering Department, University for Business and Technology, 10000 Pristina, Kosovo
12  Laboratory of Research in Biodiversity 17 Interaction Ecosystem and Biotechnology, Agronomy Department, Hydraulics Division, Faculty of Science, University 20 Août 1955, Route El Hadaik, Skikda 21000, Algeria; heddamsalim@yahoo.fr
*   Correspondence: nadhir.alansari@ltu.se (N.A.-A.); dinesh.vishwakarma4820@gmail.com (D.K.V.); Tel.: +91-967-079-1406 (D.K.V.)

**Abstract:** Nowadays, great attention has been attributed to the study of runoff and its fluctuation over space and time. There is a crucial need for a good soil and water management system to overcome the challenges of water scarcity and other natural adverse events like floods and landslides, among others. Rainfall–runoff (R-R) modeling is an appropriate approach for runoff prediction, making it possible to take preventive measures to avoid damage caused by natural hazards such as floods. In the present study, several data-driven models, namely, multiple linear regression (MLR), multiple adaptive regression splines (MARS), support vector machine (SVM), and random forest (RF), were used for rainfall–runoff prediction of the Gola watershed, located in the south-eastern part of the Uttarakhand. The rainfall–runoff model analysis was conducted using daily rainfall and runoff data for 12 years (2009 to 2020) of the Gola watershed. The first 80% of the complete data was used to train the model, and the remaining 20% was used for the testing period. The performance of the models was evaluated based on the coefficient of determination ($R^2$), root mean square error (RMSE), Nash–Sutcliffe efficiency (NSE), and percent bias (PBAIS) indices. In addition to the numerical comparison, the models were evaluated. Their performances were evaluated based on graphical plotting, i.e., time-series line diagram, scatter plot, violin plot, relative error plot, and Taylor diagram (TD). The comparison results revealed that the four heuristic methods gave higher accuracy than the MLR model. Among the machine learning models, the RF (RMSE ($m^3$/s), $R^2$, NSE, and PBIAS (%) = 6.31, 0.96, 0.94, and −0.20 during the training period, respectively, and 5.53, 0.95, 0.92, and −0.20 during the testing period, respectively) surpassed the MARS, SVM, and the MLR models in forecasting daily runoff for all cases studied. The RF model outperformed in all four models' training

and testing periods. It can be summarized that the RF model is best-in-class and delivers a strong potential for the runoff prediction of the Gola watershed.

**Keywords:** MARS; SVM; RF; rainfall; runoff; rainfall–runoff modeling

## 1. Introduction

Forecasting heavy precipitation is an important function in estimating the runoff and flooding in the short to medium term [1–4], flood warning [5], real-time flood forecasting [6], and flood mitigation [7,8]. Nonetheless, rainfall directly affects runoff generation in streams, rivers, and even floods, making it one of the most specific hydrological phenomena [2]. The socioeconomic impacts of rainfall are significant, from physical damage in floods to disruptions in transport networks [3,9]. Simultaneously, India is challenged with increasing population and climate change, which have threatened the present freshwater need for irrigation and drinking [10–13]. To overcome the challenges of water scarcity and the deterioration of cultivable land, modeling rainfall–runoff plays an important role. Many aspects of our daily lives depend on the rain we receive [14,15]. Rainfall remains one of the most influential meteorological variables [16]. The rainfall–runoff modeling in water resource management attracts many researchers and practitioners worldwide. Planning and managing water properly is the only way to prevent water stress and to balance supply and demand [17–19]. In addition to natural disasters such as floods caused by runoff from precipitation and river flow and droughts caused by short rainfall for a long duration, we can also determine the occurrence of these natural disasters by assessing the rainfall–runoff relationship [20].

The major role of several nonlinear and nonstationary variables in converting rainfall into runoff is difficult to comprehend [21]. The response to the catchment precipitation becomes more complex due to the spatiotemporal variability in rainfall intensity and uniformity [22]. However, the direct contribution of rainfall in runoff generation and runoff in streams, rivers, and even floods are one of the most focused upon hydrological phenomena. To understand the accurate relationship between the two hydrological variables, the concept of rainfall–runoff (*R-R*) plays a critical role in the area of hydrological science [23]. However, despite the remaining inconsistency in the rainfall–runoff relation, the application of machine learning is promising. These computational techniques either reduce the requirement of modeling parameters, improve modeling accuracy, or are even applicable for both purposes [24]. The main aim of this modeling is to improve our understanding of the major hydrological phenomenon, which influences all watershed systems. It also helps to develop a simulation tool to help decision makers optimize and plan the operational rules of the water resource system [25].

In rainfall–runoff modeling [2,26] and rainfall forecasting [27,28], the use of artificial intelligence (AI) and machine learning (ML) established modeling in water resources in a new direction. Several studies attempt the application of AI and ML, whether for *R-R* or for rainfall forecasting [4,9,29,30], streamflow [31–36], suspended sediment-load prediction [36–42], flood forecasting [5,6,43], stage–discharge modeling [44–48], soil temperature estimation [49–56], pan evaporation [57–68], reference evapotranspiration [69–78], soil parameter estimation such as infiltration, permeability, and saturated hydraulic conductivity [79–88], groundwater quality index [89–92], drought and stress tolerance in maize crops [93], water footprint [94,95], rice yield estimation [96], and crop coefficients [97]. Artificial neural networks (ANN) gained immense popularity in rainfall–runoff modeling [22,23,28,98] as well as rainfall forecasting [99–101], although there is no requirement for deep knowledge of hydrological processes in AI-based rainfall–runoff modeling [102]. The MLR linear mode is the most common statistical tool to predict the output–input variables. It develops a linear relationship between multiple variables [12,103,104]. A quantitative relationship exists between the dependent and independent variables in MLR [105]. The

values of the independent variables in MLR are affiliated with the values of the dependent variables [106]. The dependent, independent, and intercept variables are local behavior calculated by the least square rule or other regression rules [45].

For *R-R* modeling, AI and ML have been extensively used for decades [28]. These models have been compared to traditional statistical methods and conceptual models. The nonlinear MARS is a nonlinear and nonparametric regression [107,108]. MARS built several MLR models in the dataset range [109]. It is done by creating knots based on the splitting strategy and running a suite of the linear model for each subset; the nonlinear responses between the input and output of a dataset are divided into piecewise linear segments (splines) of different gradients [110]. The SVM is a generalized nonlinear model for both classification and regression analysis, and it was introduced by Vapnik [111]. The basic concept of SVM is to minimize the structural risk. The algorithm converts the patterns that are not linearly separable to higher-dimensional feature space using kernel functions. It attempts to reduce the upper limit of the generalization error. For its advantages over other general algorithm such as ANN, it is a better method in the hydrological field for simulation and forecasting hydrological events. The RF is a supervised ML algorithm based on bagging or bootstrap aggregation, a part of ensemble learning [112].

Al-Sudani et al. [113] hybridized the MARS model using the differential evolution algorithm (DE). They compared its performances, i.e., MARS-DE, with those of the single MARS and the least square support vector machine model (LSSVM). They reported the superiority of the hybrid MARS-DE. Adnan et al. [114] compared the performance of four ML models, i.e., the optimal pruning extreme learning machine (OPELM), MARS, M5Tree, and MARS-$K_{means}$. It was found that MARS-$K_{means}$ surpassed all other models for multi-step forecasting, i.e., one, six, and twelve hours in advance. In another study, Li et al. [115] evaluated the performances of extreme learning machines (ELM), RF, and SVM for forecasting daily, low, and peak streamflow. They reported the superiority of the ELM model.

Therefore, the present paper aims to compare the MLR, SVM, MARS, and RF models for forecasting daily runoff at the Gola watershed, located in the south-eastern part of Uttarakhand. The study was conducted with the major objectives of selecting the most relevant input variables for R-R forecasting and comparing the models' performance across the studied stations. The rest of the paper is organized as follows: Section 2 presents brief information about the study cases and data collation, Sections 3 and 4 describe the methods, and Section 5 presents the main findings and discusses their relevance in light of the literature. Finally, the main conclusions drawn from this study are given in Section 6.

## 2. Materials and Methods

### 2.1. General Description of Study Area

The Gola watershed in the south-eastern part of Uttarakhand state is shown in Figure 1. The Gola River originates in the Bhirapani valley near the village of Paharpani, Uttarakhand state, in the lesser Himalayas. The river's major tributaries are Kanchi, Kharkai, and Karkari. The watershed lies between 29°16′18″ to 29°27′33″ N latitudes and 79°46′5″ to 79°32′51″ E longitudes in northern India. The total catchment area of the Gola watershed is about 611 km². The climatic condition of the Gola watershed is mild and generally warm. The minimum and maximum elevations of the watershed are 252 m and 2302 m, respectively, above mean sea level. The Gola watershed comes under a subtropical climate with predominant seasonal rainfall. The average annual rainfall is 1699 mm, heavily influenced by monsoon rainfall. As the watershed lies on the eastern edge of the Himalayan ranges, it is subjected to heavy rainfall. It is mainly a spring-fed river; this river is a source of water for Haldwani and Kathgodam. The monsoon season extends from July to September and produces 90% of the annual rainfall. The watershed receives heavy rainfall in the months of July and August. Due to this, the mainstream of rainfed rivers like the Gola River subsequently has high discharge in these months of the year. The barrage is a landmark for

the residents, and provides irrigation water for the bhabar fields. For this reason, it is very important to know the daily forecasting of the river flow to avoid any risk/distress/fatality.

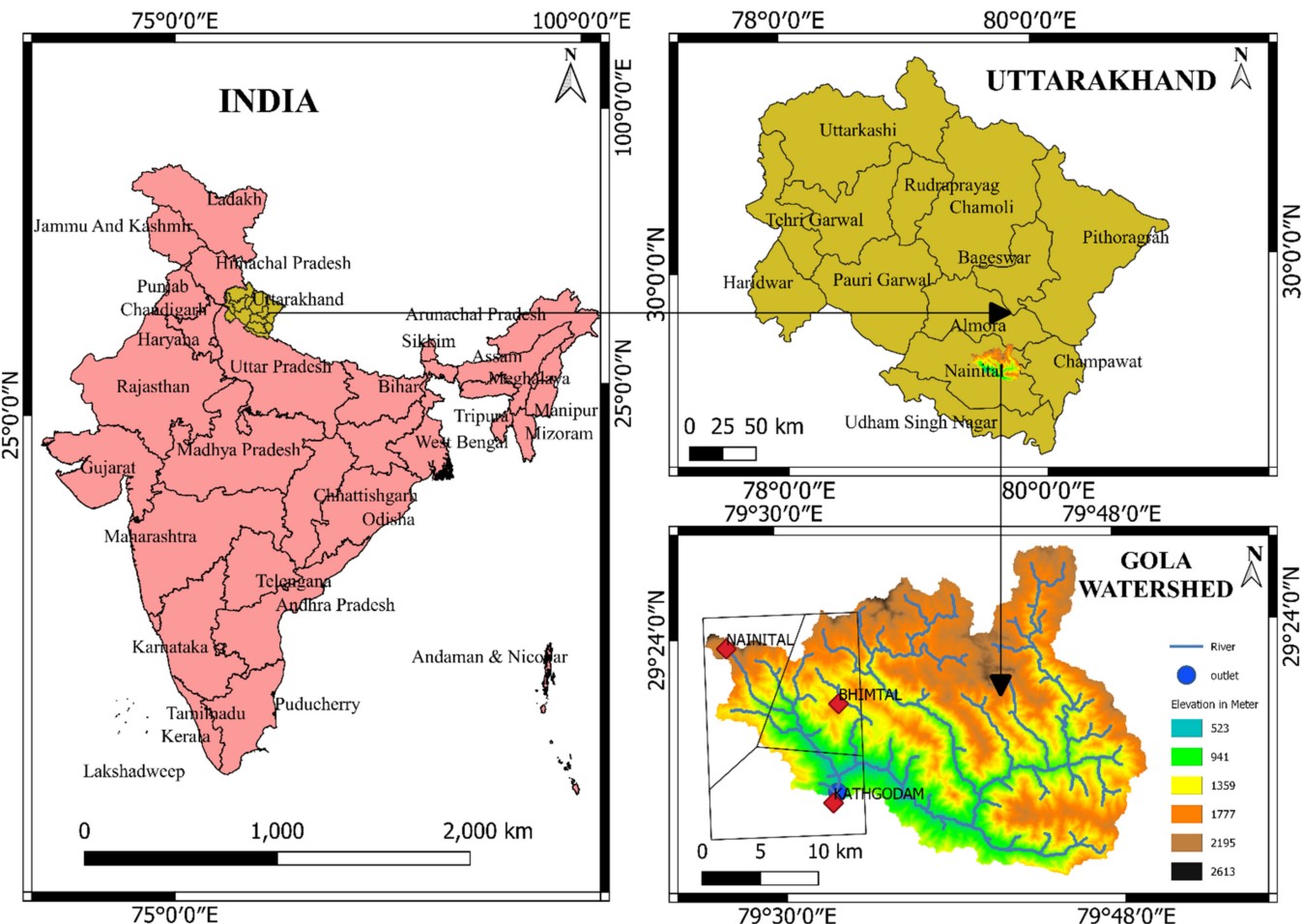

**Figure 1.** Location map of the study area.

## 2.2. Data Acquisition and Input Data Preparation

The daily data of rainfall and runoff for 12 years (2009 to 2020) of the study location (Gola watershed) were used to analyze rainfall–runoff modeling. The runoff data of the Gola River were taken from an observation station at Kathgodam barrage. The rainfall data of three rain-gauge stations—Nainital, Bhimtal, and Kathgodam—were taken from the respective irrigation departments (Figure 2a). The Thiessen polygon method was used to calculate the mean areal rainfall of the Gola watershed. Plots of the rainfall and runoff time-series data are shown in Figure 2b,c, respectively. The daily rainfall and runoff data for 12 years were used to develop and validate the models.

Statistical parameters were used to analyze the time-series dataset for rainfall–runoff modeling of the Gola watershed and are presented in Table 1. The complete dataset was divided into training and testing datasets. The first 80% of the complete data was used in training, and the remaining 20% was used for the testing period. During the division of the datasets into training and testing subsets, cross-validation of the dataset is necessary to obtain the same statistical population. The skewness value of the dataset showed that the distribution was highly skewed. Figure 3 shows the flowchart of the proposed methodology.

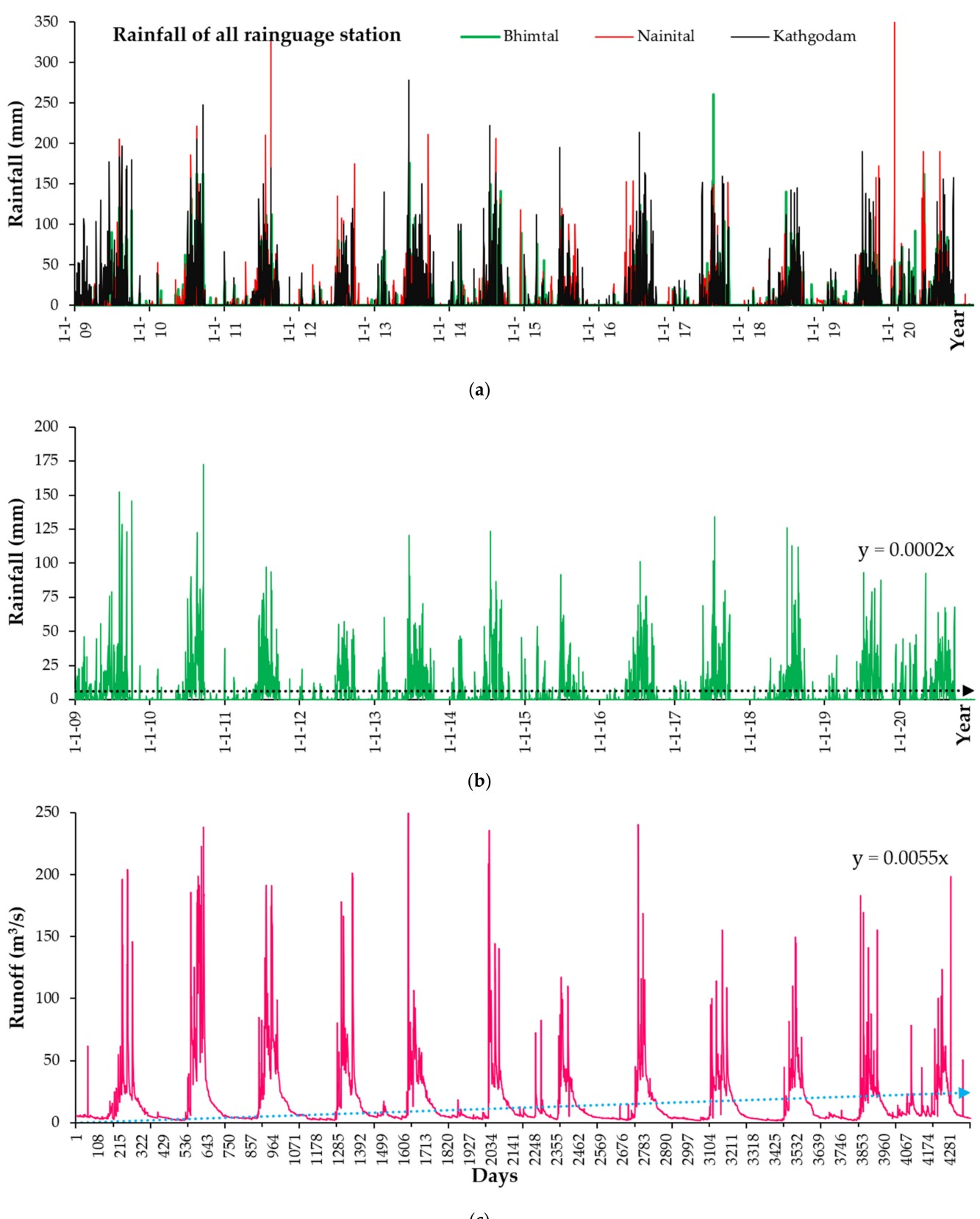

**Figure 2.** (**a**) Rainfall data from three raingauge stations, namely, Bhimtal, Nainital and Kathgodam; (**b**) mean areal rainfall time-series data for Gola watershed and (**c**) runoff time-series data for Gola watershed.

**Table 1.** Basic statistics of training, testing, and total rainfall and runoff datasets at study stations.

| Statistical Parameters | Mean | Median | Minimum | Maximum | Standard Deviation | CV (%) | Skewness |
|---|---|---|---|---|---|---|---|
| Total dataset | | | | | | | |
| Rainfall (mm) | 6.45 | 0 | 0 | 172.38 | 15.60 | 24.18 | 3.87 |
| Runoff (m$^3$/s) | 17.22 | 6.06 | 1.88 | 250.03 | 27.51 | 15.97 | 3.80 |
| Training data | | | | | | | |
| Rainfall (mm) | 6.45 | 0 | 0 | 172.38 | 15.87 | 24.60 | 4.02 |
| Runoff (m$^3$/s) | 17.35 | 5.66 | 1.38 | 250.03 | 28.69 | 16.54 | 3.76 |
| Testing data | | | | | | | |
| Rainfall (mm) | 6.89 | 0 | 0 | 111.69 | 14.47 | 21.00 | 3.07 |
| Runoff (m$^3$/s) | 16.83 | 7.5 | 1.61 | 197.08 | 22.16 | 13.16 | 3.59 |

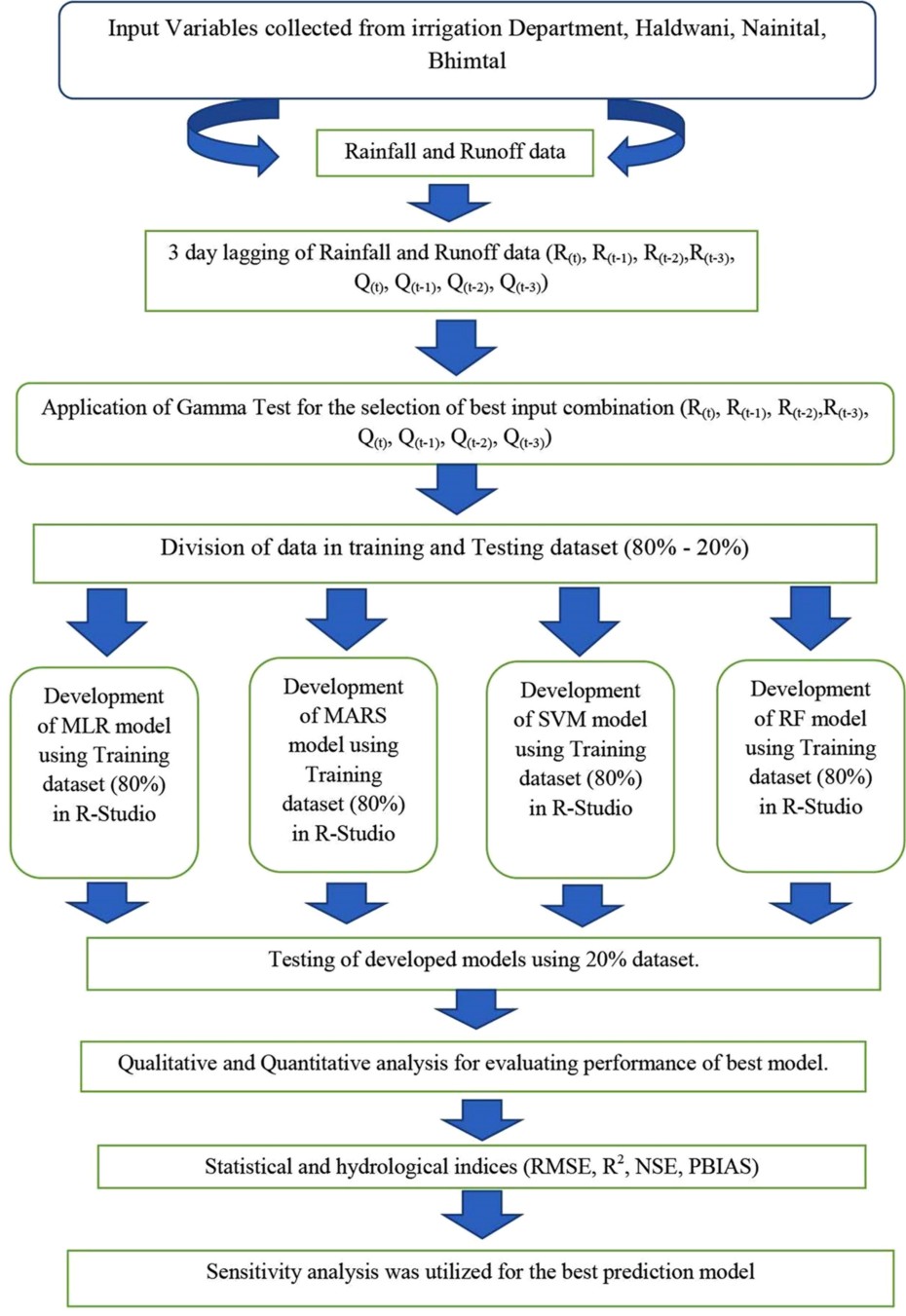

**Figure 3.** Flowchart of methodology.

### 2.3. Gamma Test

The Gamma test selects the best input variables in modeling a dataset [47,116–119]. It is a flexible and unbiased tool for evaluating the potential of each input parameter. Traditionally, trial and error methods were used to select the best input variable, making it a time-consuming and tedious job that includes training and testing every possible input combination to select the best suitable input vector. It also fails to provide information about the number of data points necessary for calibration to achieve the accuracy of the optimum model. The Gamma test plays a significant role by guiding the selection of various input parameters to develop reliable and smooth models. Nonlinear correlation between random variables is evaluated by Gamma tests, like the nonlinear correlation between the input and output pairs. The idea of the Gamma test was first discovered by Stefansson et al. [120] for simulation; it was then another researcher who further adapted it for research activity [38,47,121–123]. It was used for estimating the minimum standard error for nonlinear models for any input variables [121].

The mathematical Gamma test is represented as:

$$\{X_1(i), \ldots \ldots \ldots X_n(i), Y_i\} = \{(X_i, Y_i)|1 \leq i \leq N\} \tag{1}$$

where X is the $X_1, \ldots, X_n$ corresponds to the predictor's variables, i.e., m variables for a total of N data points, scalar $Y_i$ is the output variable, and Gamma ($\Gamma$) is calculated by building up a linear regression between input (X) and output (Y), as:

$$Y = f(X_i, \ldots \ldots \ldots X_n) + \Gamma \tag{2}$$

where f is a smooth function and $\Gamma$ corresponds to the noise. The overall model complexity is evaluated according to Equation (2). More suitable input variables were reflected by a low value of $\Gamma$, i.e., close to zero. In addition, a complicated model is obtained based on the obtained gradient value, and based on the standard error (SE) of $\Gamma$, a more reliable Gamma value is obtained. In addition, $V_{ratio}$, given by Equation (3), indicates the predictability of the output variables. A model's complexity can be determined from the output Y of Equation (2). A value of $\Gamma$ close to zero indicates a suitable input variable. We have a complicated model when the gradient is high; we have a simple model if the gradient is low. The Gamma value is more reliable if the standard error (SE) of $\Gamma$ is smaller. $V_{ratio}$, given in Equation (3), measures the predictability of a variable.

$$V_{ratio} = \frac{\Gamma}{\sigma^2(Y)} \tag{3}$$

where $\sigma^2(y)$ is the output variance of Y, and $\Gamma$ is the Gamma function. When Vratio is near 0, the predictability is higher. We can build a more qualitative mathematical model with smaller values for Gamma ($\Gamma$), gradient, SE, and $V_{ratio}$.

## 3. Software Application Used in the Study

The machine learning techniques, MLR, MARS, SVM, and RF, were used in the rainfall–runoff modeling of the Gola watershed. The description of these models is as follows.

### 3.1. Multiple Linear Regression

Multiple linear regression is the simplest statistical technique to predict the output from several input variables. The linear relationship is developed between multiple variables. In this regression, a quantitative relationship is made by independent and dependent variables. The values of independent variables are related to dependent variables [106].

MLR predicts runoff from input rainfall by taking the dataset into training and testing data periods. The expression for MLR is defined as follows:

$$Y = \alpha_0 + \alpha_1 X_1 + \alpha_2 X_2 + \alpha_3 X_3 \ldots \ldots \ldots + \alpha_n X_n \tag{4}$$

where Y = the output or the modeled variable; $X_1$, $X_2$ ... ... . $X_n$ = the inputs variables; $\alpha_0$ = intercept, and $\alpha_1$, $\alpha_2$ ... ... $\alpha_n$ = regression coefficients.

### 3.2. Multivariate Adaptive Regression Splines (MARS)

Multivariate adaptive regression splines (MARS) is a nonlinear and nonparametric regression method that built several MLR models across the range of predictor values. It is done by splitting the data and running a linear model on each different partition. The nonlinear responses between the input and output of a dataset are divided into piecewise linear segments (splines) of different gradients [124]. The extensible regression models proposed by MARS have the solution space for each model divided into intervals, and splines fit every interval space [125]. There is the creation of a bias function and finding a potential knot location to improve the model's performance and over-fit. The backstage of the MARS model is done by pruning the ineffective term [126]. The comparison of different model subsets is done by the less expansive technique of generalized cross-validation [127]. It is expressed as follows:

$$GCV = \frac{MSE}{\left[1 - \frac{(f+1)+pf}{n}\right]^2} \tag{5}$$

where $MSE$ = mean square error; $f$ = number of bias functions; $p$ = bias function penalty; and $n$ = number of observations.

The MARS model performs under two types of forward and backward functions [126]. In the forward function stage, the model develops a huge quantity of bias functions introduced by the MARS model. The generalized form of the MARS model is given below [128]:

$$Y = \beta_o + \sum_{i=1}^{m} \beta_i H_{ei}(X_v(e,\ I)) \tag{6}$$

where Y = output parameter; $\beta_o$ = constant value; I = number of bias functions; $H_{ei}(X_v(e,I))$ = ith bias function; and $\beta_i$ = the corresponding coefficient of $H_{ei}(X_v(e,\ I))$.

The model has a collection of bias functions. In the second stage, it can estimate the least square model. The MARS model is defined as follows [129]:

$$Y = \partial_o + \sum_{i=1}^{n} \partial_i h_i(X) \tag{7}$$

where $h_i(X)$ = the splines function; $\partial$ = the coefficient of the spline function; and i = the total number of functions in the model.

### 3.3. Support Vector Machine (SVM)

The support vector machine (SVM) is a supervised ML model that uses a nonlinear generalization algorithm to classify two groups and regression problems. The foundation of the SVM was made by Vapnik [111,130], and Bray and Han [131] introduced it. SVMs are generalized linear classifiers and supervised learning methods for regression and data classification. The kernel allows SVMs to form nonlinear boundaries. The different kernel functions include linear kernel, polynomial kernel, radial basis, and sigmoid kernel. The algorithm's expression by the dataset's inner products is called the dual problem. Support vector regression (SVR) was developed by Vladimir Vapnik [111]. It is characterized by using kernels, sparse solution, and Vapnik–Chervonenkis (VC) controls of margins and many support vectors.

SVR is a powerful tool in real-value function estimation. It estimates continuous-value multivariate functions. It uses a supervised learning approach: SVR trains by taking symmetrical loss functions and reducing high and low misestimation [132]. Support vector regression (SVR) attempts to minimize the upper limit of the generalization error instead of fixing the empirical error. The first formulation of SVR is a hard-margin solution that contributes to overfitting. The soft margin appears to generalize in the presence of outliers

and noise. It prevents overfitting, which makes it favorable for forecasting research work. It has high generalization capability and great prediction accuracy. SVMs formulate binary classification problems to convex optimization problems [111]. The $\varepsilon$-intense region around the function allows SVM generalization to SVR, also called $\varepsilon$-tube. It helps reformulate the optimization problem in continuous-value functions, which helps to balance model complexity and prediction error. Considering the training dataset, T, represented as:

$$T = \{(x_1, y_1), (x_2, y_2)\cdots\cdots(x_m, y_m)\} \tag{8}$$

where $x \in X\ R^n$ are the training inputs and $y \in Y \subset R^n$ are the training outputs. Assume a nonlinear function y is given by:

$$f(x) = \mathbf{w^T\Phi(x_i)} + b \tag{9}$$

where *f(x)* = nonlinear function; T = training data; w = weight vector, *b* = bias; and $\mathbf{\Phi}$ ($x_i$) = higher dimensional feature space by the linear mapping function of input space x. The main objective is to fit the dataset T with the help of function *f(x)*, having the highest deviation $\varepsilon$ from the training dataset T. The equation is now transformed into a constrained complex problem, as follows:

$$\min. \left(\frac{1}{2}\mathrm{w^Tw}\right) \text{subject to}: \begin{cases} \mathrm{yi} - \left(\mathbf{w^T\Phi(x_i)} + b\right) \leq \varepsilon \\ \mathrm{yi} - \left(\mathbf{w^T\Phi(x_i)} + b\right) \geq \varepsilon \end{cases} \tag{10}$$

where $\varepsilon$ ($\geq$0) is the maximum acceptable deviation. Equation (7) can be written as:

$$\min. \left(\frac{1}{2}\mathrm{w^Tw}\right) \text{subject to}: \begin{cases} \mathrm{yi} - \mathbf{w^T\Phi(x_i)} - b \leq \varepsilon \\ \mathbf{w^T\Phi(x_i)} + b - \mathrm{y}_i \leq \varepsilon \end{cases} \tag{11}$$

Further, the final expression for SVM becomes:

$$f(x) = \sum_{i=1}^{m} \left(\alpha_i^+ - \alpha_i^-\right) K\left(x_i, x_j\right) + b \tag{12}$$

where $\alpha_i^+$ and $\alpha_i^-$ are the Langrangian multipliers, and $K\left(x_i, x_j\right)$ is the kernel function. [133].

*3.4. Random Forest (RF)*

Random forest is a supervised ML that uses ensemble learning techniques for classification and regression problems. It is a technique that predicts from different machine learning algorithms or the same algorithm several times for more accurate predictions. RF uses the bagging technique or bootstrap aggregation, part of ensemble learning. It generates several subsets of data from training samples chosen randomly with replacement. Each subset of data is used to train its decision tree.

The bootstrap aggregation technique reduces the variance of an estimated prediction function. Bagging works excellently for high variance and low bias, such as decision trees. Random forest constructs multiple decision trees during training. It combines the prediction results from each decision tree to give the final output. Decision trees are computationally expansive; these are very sensitive to data on which they are trained and may experience deviation in predictions if the underlying data are changed. Several decision trees are constructed by the algorithm that operates the model. RF is the aggression of tree predictors. The trees are estimated by the values of a random vector computed from the same distribution for each forest tree [134]. In the RF model, every tree is grown with a random subset of variables [135]. It ensures that the bagged trees are in the way that a single tree reduces the correlation and variance between trees of the model. Each decision tree picks a random sample from the dataset, while generating its split adds a further element of randomness to minimize the problem of overfitting. The random forest chooses nodes from a subset of available features that breaks variables at each node to

reduce the association between the trees. The mean square error (*MSE*) can be calculated as [127]:

$$MSE = \frac{1}{N} \sum_{i=1}^{m} (Z_i - i)^2 \tag{13}$$

where $Z_i$ = the measured variable value and $i$ = the mean of all out-of-bag (OOB) predictions.

The RF model comes under the classification of regression tree (CART) tools and is used for classification and regression problems. Many RF trees are key parameters; the model performance can be evaluated by out-of-bag (OOB). Random forest can help over-fitting the model for the training dataset, which can be evaded by selecting input data during the training cycle and establishing variation in weak learners [136]. The RF model makes multiple decision trees, and the output of the models can be estimated by taking the mean output of every tree. The predicted values are calculated as:

$$Y = \frac{1}{N} \sum_{i=1}^{N} R(x) \tag{14}$$

where $Y$ = the predicted output by the RF model, $N$ = the number of trees (*n*-tree) utilized in the RF model, and $R(x)$ = the results of every random tree.

## 4. Performance Evaluation of Models

The performances of the MLR, MARS, SVM, and RF models were evaluated based on the coefficient of determination ($R^2$), root mean square error (*RMSE*), Nash–Sutcliffe efficiency (*NSE*), and percent bias (*PBIAS*), and visual interpretation using a line diagram, scatter diagram, violin plot, and relative and Taylor diagrams. The $R^2$, *RMSE* [12,47,117,137,138], *NSE* [139], and *BIAS* [119,140,141] are described as:

$$R^2 = \left[ \frac{\left[ \sum Q_o Q_p \right] - \left[ \frac{\sum Q_o \sum Q_p}{N} \right]}{\sqrt{\left[ \sum Q_p{}^2 - \frac{(\sum Q_p)^2}{N} \right] \left[ \sum Q_o{}^2 - \frac{(\sum Q_o)^2}{N} \right]}} \right]^2 \tag{15}$$

$$RMSE = \sqrt{\frac{1}{N} \sum_{i=1}^{N} (Q_o - Q_p)^2} \tag{16}$$

$$NSE = 1 - \left[ \frac{\sum_{i=1}^{N} (Q_o - Q_p)^2}{\sum_{i=1}^{N} (Q_o - \overline{Q_o})^2} \right] \tag{17}$$

$$PBIAS = \left[ \frac{\sum_{i=1}^{N} (Q_o - Q_p)}{\sum_{i=1}^{N} (Q_o)} \right] \times 100 \tag{18}$$

where $Q_o$ = the observed runoff value; $Q_p$ = the predicted runoff value; $N$ = the total number of values of the variable in the dataset; and $\overline{Q_o}$ = the mean of the observed discharge data.

The coefficient of determination describes the statical relationship (collinearity) between the variables and helps to show the nature of association among the predicted and observed data. $R^2$ is the ratio of explained variation compared to the total variation [142]. The coefficient of multiple determination measures the percentage of various independent variables, which can be explained by variations in the independent variables when taken together [143]. It ranges from 0 to 1; its higher value indicates less error variance, and generally, a value greater than 0.5 is considered acceptable [144,145]. It is famously used in model evaluation. This statistical tool is highly sensitive to outliers and insensitive to additive and proportional differences between observed and predicted data [146]. The square root of the average square of all of the errors is called root mean square error (*RMSE*) [104]. It is an excellent general-purpose error matrix commonly used for the numerical prediction

model. *RMSE* has a good measure of accuracy, but it can only compare the prediction error of models or configure a particular variable and not between two different variables, making it scalar-dependent.

*RMSE* lies between 0 to ∞ [47]. NSE is a normalized statistical tool determining the relative magnitude of residual variance or noise. NSE lies between −∞ and 1 and is less sensitive to high extreme values [139]. Percent bias measures the relationship between the observed data and its predicted data; it measures the average tendency of observed data to be larger or smaller than the predicted data [147]. Percent bias describes whether the simulated model is overestimated or underestimated. A low PBIAS value or a value that tends to zero indicates the optimal model. A negative value indicates the overestimation of the model.

In contrast, a positive PBIAS value indicates an underestimation of the model [119, 140,141,147]. When the data are evaluated, PBIAS reveals any deviation of the data as a percentage [148]. A model with higher $R^2$ and NSE values, and lower RMSE and PBAIS values, decrees a relatively better model for the simulation of $Q_t$.

## 5. Results and Discussion

This section deals with the development and results of runoff prediction models using ML techniques for the Gola watershed. This study categorizes runoff modeling into two approaches based on the machine learning model context. One approach considers that a forecasting model can be based on the river flow data by including the correlated lag times as an attribute to forecast one step. On the other hand, the second approach to modeling river flow entails an appropriate exogenous hydrological variable apart from the river flow data. Multiple linear regression (MLR), multivariate adaptive regression splines (MARS), support vector machine (SVM), and random forest (RF) models were applied to develop the rainfall–runoff models. The qualitative performance evaluation of the models was achieved by visual observations such as time series, scatterplot, violin plot, relative error, and Taylor plot, and quantitative evaluations were carried out using different statistical and hydrological performance indices, namely, root mean square error (RMSE), coefficient of determination ($R^2$), the Nash–Sutcliffe coefficient of efficiency (NSE), and percent bias (PBIAS).

### 5.1. Selection of Best Input Combination

Selecting the most appropriate input variables is vital to model development [79]. In the present investigation, the GT algorithm was used for selecting the relevant input variable combination for runoff prediction. In this study, various combinations of present-day runoff ($Q_{(t)}$), previous day runoff ($Q_{(t-1)}$), previous two days' runoff ($Q_{(t-2)}$), previous three days' runoff ($Q_{(t-3)}$), present-day rainfall ($R_{(t)}$), previous day rainfall ($R_{(t-1)}$), previous two days' rainfall ($R_{(t-2)}$) and previous three days' rainfall ($R_{(t-3)}$) were used for Gamma testing (Table 2). The models with low Gamma ($\Gamma$) and $V_{ratio}$ values were considered the most appropriate for developing the models [149]. It was noticed that the Gamma value and $V_{ratio}$ decreased with an increase in the number of predictors. However, after a certain point, the Gamma value began to increase again. This might be due to the following two reasons: (i) the inclusion of a high number of input variables may be the cause of overfitting, and (ii) the inclusion of a smaller number of input variables results in the incapacity of the model to correctly explain the total variance of the forecasted subset. The minimum Gamma ($\Gamma$) and $V_{ratio}$ values were 0.407 and 0.191, respectively, for the M19 predictor set. Hence, the M19 predictor combination was employed for further analysis. It could be stated that using rainfall with a two-day lag and the discharge from one to three days' lag as a predictor would produce an optimum rainfall–runoff model. It was also noticed that the Gamma value and $V_{ratio}$ increased when the rainfall of the three-day lag was included in the predictors. This might be due to the low correlation of the predictor variable with the predictand.

**Table 2.** Gamma statistics for different input combinations.

| Model No. | Model Input Combination | Mask | Gamma | V-Ratio |
|---|---|---|---|---|
| M1 | $Q_{(t-1)}$ | 0000001 | 0.082 | 0.329 |
| M2 | $R_{(t)}, Q_{(t-1)}$ | 1000001 | 0.061 | 0.244 |
| M3 | $R_{(t)}, R_{(t-1)}, Q_{(t-1)}$ | 1100001 | 0.064 | 0.256 |
| M4 | $R_{(t)}, R_{(t-1)}, R_{(t-2)}, Q_{(t-1)}$ | 1110001 | 0.056 | 0.225 |
| M5 | $R_{(t)}, R_{(t-1)}, R_{(t-2)}, R_{(t-3)}, Q_{(t-1)}$ | 1111001 | 0.051 | 0.207 |
| M6 | $R_{(t)}, R_{(t-1)}, R_{(t-2)}, R_{(t-3)}, Q_{(t-3)}, Q_{(t-1)}$ | 1111101 | 0.054 | 0.217 |
| M7 | $Q_{(t-1)}, Q_{(t-2)}$ | 0000011 | 0.081 | 0.320 |
| M8 | $R_{(t)}, Q_{(t-1)}, Q_{(t-2)}$ | 1000011 | 0.063 | 0.254 |
| M9 | $R_{(t)}, R_{(t-1)}, Q_{(t-1)}, Q_{(t-2)}$ | 1100011 | 0.051 | 0.212 |
| M10 | $Q_{(t-1)}, Q_{(t-2)}, Q_{(t-3)}$ | 0000111 | 0.076 | 0.307 |
| M11 | $R_{(t)}, R_{(t-1)}, Q_{(t-1)}, Q_{(t-2)}, Q_{(t-3)}$ | 1000111 | 0.053 | 0.213 |
| M12 | $R_{(t)}, R_{(t-2)}, Q_{(t-1)}, Q_{(t-2)}, Q_{(t-3)}$ | 1100111 | 0.048 | 0.194 |
| M13 | $R_{(t)}, R_{(t-1)}, R_{(t-2)}, R_{(t-3)}$ | 1111000 | 0.107 | 0.430 |
| M14 | $R_{(t)}, R_{(t-1)}, R_{(t-2)}, R_{(t-3)}, Q_{(t-1)}$ | 1111001 | 0.061 | 0.207 |
| M15 | $R_{(t)}, R_{(t-1)}, R_{(t-2)}, Q_{(t-1)}, Q_{(t-2)}, Q_{(t-3)}$ | 1110111 | 0.050 | 0.200 |
| M16 | $R_{(t)}, R_{(t-1)}, R_{(t-2)}$ | 1110000 | 0.114 | 0.459 |
| M17 | $R_{(t)}, R_{(t-1)}, R_{(t-2)}, Q_{(t-1)}$ | 1110001 | 0.056 | 0.225 |
| M18 | $R_{(t)}, R_{(t-1)}, R_{(t-2)}, Q_{(t-1)}, Q_{(t-2)}$ | 1110011 | 0.052 | 0.209 |
| M19 | **$R_{(t)}, R_{(t-1)}, R_{(t-2)}, Q_{(t-1)}, Q_{(t-2)}, Q_{(t-3)}$** | 1110111 | 0.047 | 0.191 |
| M20 | $R_{(t)}, R_{(t-1)}$ | 1100000 | 0.124 | 0.498 |
| M21 | $R_{(t)}, R_{(t-1)}, Q_{(t-1)}$ | 1100001 | 0.064 | 0.256 |
| M22 | $R_{(t)}, R_{(t-1)}, Q_{(t-1)}, Q_{(t-2)}$ | 1100011 | 0.053 | 0.212 |
| M23 | $R_{(t)}, R_{(t-1)}, Q_{(t-1)}, Q_{(t-2)}, Q_{(t-3)}$ | 1100111 | 0.073 | 0.194 |
| M24 | $R_{(t)}, R_{(t-1)}, R_{(t-3)}, Q_{(t-1)}, Q_{(t-2)}, Q_{(t-3)}$ | 1101111 | 0.050 | 0.203 |
| M25 | $R_{(t)}, Q_{(t-1)}$ | 1000001 | 0.061 | 0.244 |
| M26 | $R_{(t)}, Q_{(t-1)}, Q_{(t-2)}$ | 1000011 | 0.063 | 0.254 |
| M27 | $R_{(t)}, Q_{(t-1)}, Q_{(t-2)}, Q_{(t-3)}$ | 1000111 | 0.053 | 0.213 |
| M28 | $R_{(t)}, R_{(t-1)}, Q_{(t-1)}, Q_{(t-2)}, Q_{(t-3)}$ | 1100111 | 0.048 | 0.194 |
| M29 | $R_{(t)}, R_{(t-1)}, R_{(t-2)}, R_{(t-3)}$ | 1111000 | 0.107 | 0.430 |
| M30 | $R_{(t)}, R_{(t-1)}, R_{(t-2)}, R_{(t-3)}, Q_{(t-1)}$ | 1111001 | 0.051 | 0.207 |
| M31 | $R_{(t)}, R_{(t-1)}, R_{(t-2)}, Q_{(t-1)}, Q_{(t-2)}$ | 1110011 | 0.052 | 0.209 |
| M32 | $R_{(t)}, R_{(t-1)}, R_{(t-2)}, R_{(t-3)}, Q_{(t-1)}, Q_{(t-2)}$ | 1111011 | 0.050 | 0.200 |
| M33 | $R_{(t)}, R_{(t-1)}, R_{(t-2)}, R_{(t-3)}, Q_{(t-1)}, Q_{(t-2)}, Q_{(t-3)}$ | 1111111 | 0.048 | 0.194 |

### 5.2. *Application of Machine Learning Techniques for Rainfall–Runoff Modeling*

As per the GT result, the best input combination for the development of the MLR, MARS, SVM, and RF models was made based on the following equation:

$$Q_{(t)} = f\left(R_{(t)}, R_{(t-1)}, R_{(t-2)}, Q_{(t-1)}, Q_{(t-2)}, Q_{(t-3)}\right) \tag{19}$$

#### 5.2.1. MLR Model for Runoff Prediction

The MLR technique was used to predict the runoff of the Gola watershed using the best input combination based on the Gamma test results in the R-Studio environment. The developed MLR model with its training dataset can be formulated as follows:

$$Q_{(t)} = 1.11 + 0.55R_{(t)} + 0.01R_{(t-1)} - 0.03R_{(t-2)} + 0.48Q_{(t-1)} + 0.21Q_{(t-2)} + 0.03Q_{(t-3)} \tag{20}$$

The qualitative performance assessment of the MLR model for predicting the runoff of the Gola watershed was done using a graphical comparison between the ordinates of the observed and predicted runoff values (Figures 4 and 5). The visual observation revealed that there was a large variation in the observed and predicted peak values of the runoff. It was also observed that the MLR model underpredicts higher flow/runoff and overpredicts lower flow/runoff in both the training and testing periods (Figure 4a,b). The MLR, being a linear model, could not capture the nonlinearity in the predictor–predictand relationship. Hence, the MLR model explained the medium range of variance in the predictand variable

better than the extreme values. In other words, these models simulated the average runoff values more effectively than extreme events. The average points are well simulated in most cases.

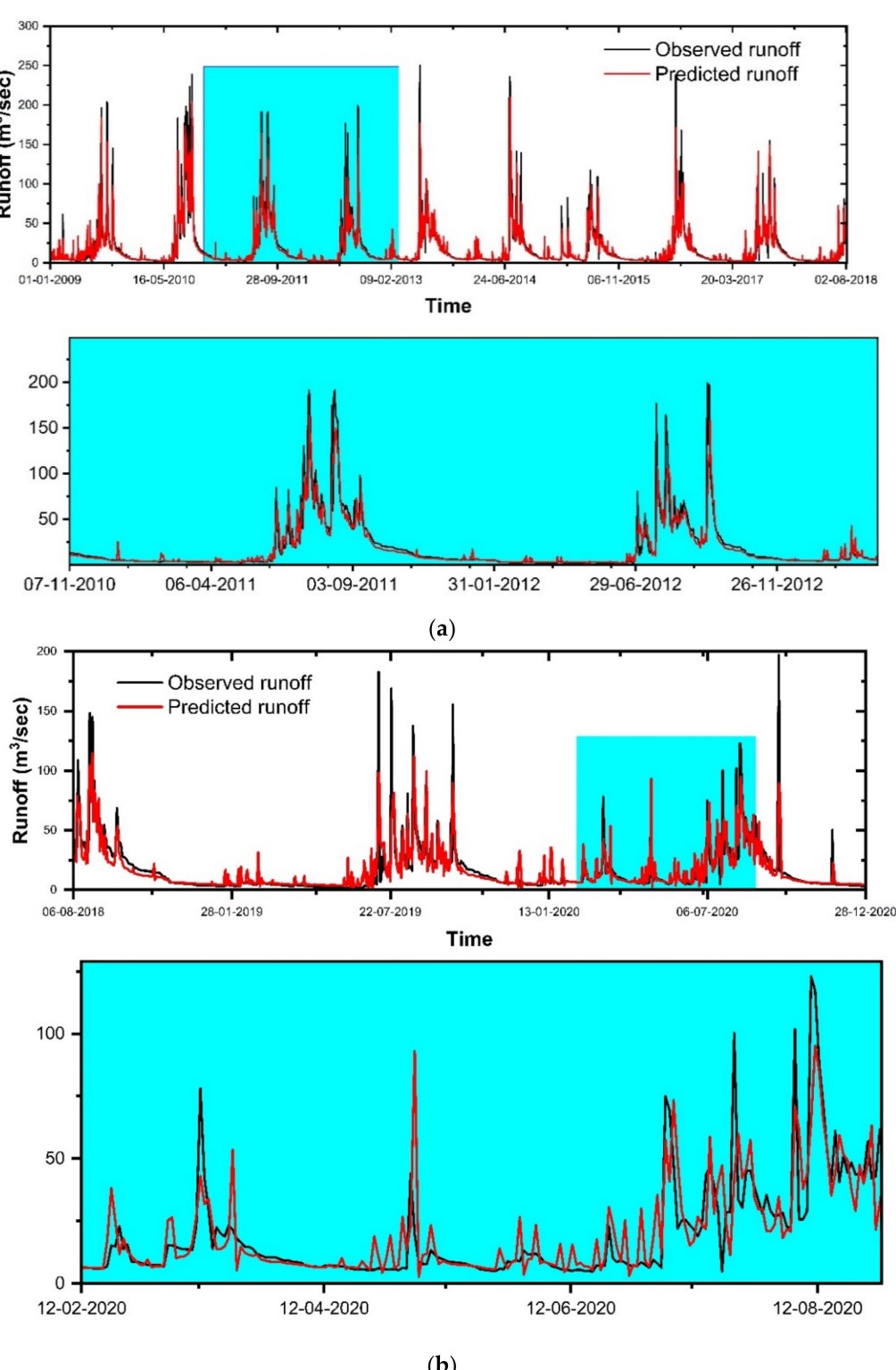

**Figure 4.** Results of simulation of the runoff (m$^3$/s) of Gola watershed using MLR model from 2009 to 2020 (**a**) training and (**b**) testing period.

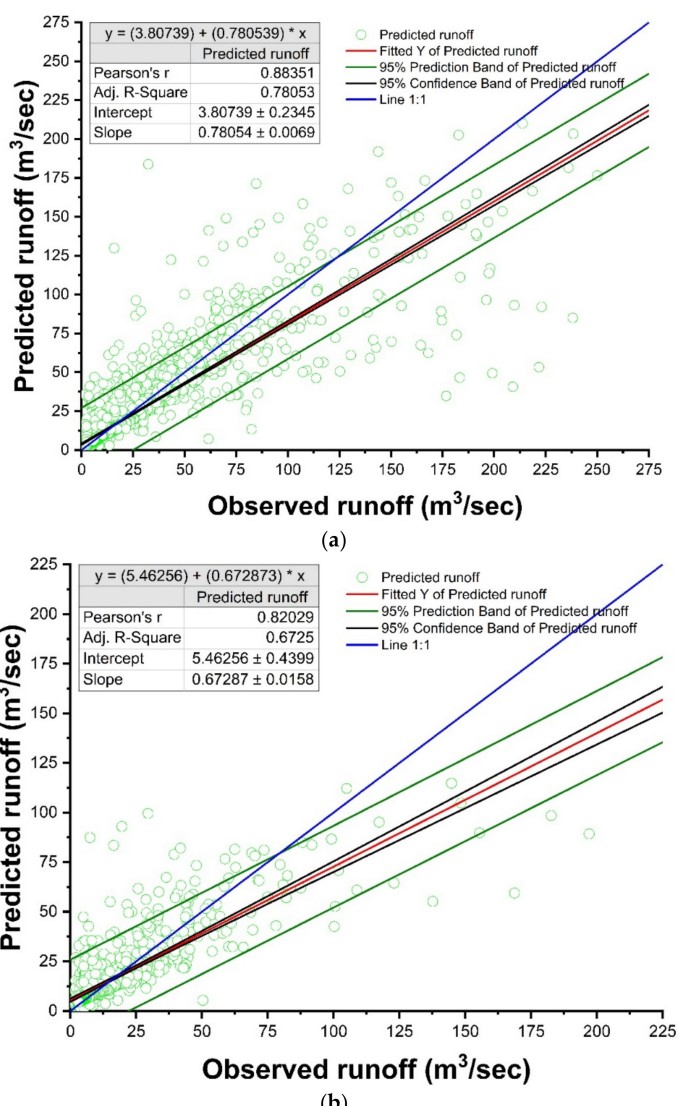

**Figure 5.** Scatter plots, evaluation statistics, and confidence intervals of the observed and predicted daily runoff Q(t) using MLR model. (**a**) Training dataset and (**b**) testing dataset.

The values of RMSE, $R^2$, NSE, and PBIAS were 13.44 m$^3$/s, 0.78, 0.72, and 0.00%, respectively, during the training and 12.67 m$^3$/s, 0.67, 0.51, and 0.80%, respectively, during the testing period for the MLR model (Table 3). The model revealed low training bias and underestimated runoff values in the testing period. It was seen that the MLR model lacked the satisfactory mapping of the Gola watershed's runoff. According to Figure 5a,b, using 95% confidence intervals, the results showed that most of the points of simulated runoff values (m$^3$/s) are outside of the confidence range, which indicates overestimation and underestimation of the target points in both periods. However, according to the presented results, the model's performance is not acceptable.

**Table 3.** Comparison of different machine learning models for daily runoff prediction.

| Model | Training | | | | Testing | | | |
|---|---|---|---|---|---|---|---|---|
| | RMSE (m$^3$/s) | $R^2$ | NSE | PBIAS (%) | RMSE (m$^3$/s) | $R^2$ | NSE | PBIAS (%) |
| MLR | 13.44 | 0.78 | 0.72 | 0 | 12.67 | 0.67 | 0.51 | 0.80 |
| MARS | 12.55 | 0.81 | 0.76 | 0 | 10.07 | 0.79 | 0.74 | 0.20 |
| SVM | 12.62 | 0.83 | 0.81 | 4.10 | 14.02 | 0.60 | 0.60 | −0.40 |
| RF | 6.31 | 0.96 | 0.94 | −0.20 | 5.53 | 0.95 | 0.92 | −0.20 |

### 5.2.2. MARS Model for Runoff Prediction

In the case of MARS modeling, the backward pass was used to prune the model by deleting unnecessary bias functions at every stage until the supermodel was found and good generalization ability was achieved. The values of the GCV parameter for best models were 159.80 and 107.05 for the training and testing sets, respectively. RMSE, $R^2$, NSE, and PBIAS were 12.55 m$^3$/s, 0.81, 0.76, and 0.00%, respectively, during training, and 10.07 m$^3$/s, 0.79, 0.74, and 0.20%, respectively, during the testing period for the MARS model (Table 4). The temporal variations and scatter plots of the observed and predicted runoff during the training and testing period are displayed in Figures 6 and 7, respectively. As can be seen from Figure 6a,b, there is good agreement between the observed runoff values and the corresponding values simulated by the MARS model in the training and testing periods, respectively. The trend of predicting runoff was satisfactory for the observed runoff of the Gola watershed. The peak values of the runoff were not predicted with great accuracy. Low values of PBIAS were found in the MARS model during the training period, which indicates an accurate model simulation. The PBIAS value of 0.00% and the positive value of 0.20% indicated slight underestimation during the testing period. According to Figure 7a,b, using 95% confidence intervals, the results showed that some of the points of the simulated runoff values (m$^3$/s) are outside of the confidence range, which indicates the overestimation and underestimation of the target points in both periods. However, the model's performance is acceptable according to the presented results.

**Table 4.** Results of different performance indicators for RF models during training and testing sets.

| Models | Training | | Testing | |
|---|---|---|---|---|
| | RMSE | $R^2$ | RMSE | $R^2$ |
| RF-1 | 6.443 | 0.95 | 5.553 | 0.95 |
| RF-2 | 6.388 | 0.95 | 5.576 | 0.95 |
| RF-3 | 6.351 | 0.96 | 5.572 | 0.94 |
| RF-4 | 6.423 | 0.95 | 5.550 | 0.95 |
| RF-5 | 6.442 | 0.95 | 5.621 | 0.94 |
| RF-6 | 6.480 | 0.95 | 5.459 | 0.95 |
| RF-7 | 6.415 | 0.95 | 5.609 | 0.95 |
| RF-8 | 6.371 | 0.95 | 5.677 | 0.94 |
| RF-9 | 6.404 | 0.95 | 5.598 | 0.95 |
| RF-10 | 6.378 | 0.95 | 5.521 | 0.95 |
| RF-11 | 6.403 | 0.95 | 5.430 | 0.95 |
| RF-12 | 6.445 | 0.95 | 5.481 | 0.95 |
| RF-13 | 6.373 | 0.95 | 5.563 | 0.95 |
| RF-14 | 6.449 | 0.95 | 5.500 | 0.95 |
| RF-15 | 6.447 | 0.95 | 5.468 | 0.95 |
| RF-16 | 6.435 | 0.96 | 5.571 | 0.94 |
| RF-17 | 6.386 | 0.95 | 5.580 | 0.95 |
| RF-18 | 6.395 | 0.95 | 5.539 | 0.95 |
| RF-19 | 6.481 | 0.95 | 5.451 | 0.95 |
| RF-20 | 6.408 | 0.95 | 5.575 | 0.95 |
| RF-21 | 6.375 | 0.95 | 5.626 | 0.94 |
| RF-22 | 6.389 | 0.95 | 5.529 | 0.95 |
| RF-23 | 6.451 | 0.95 | 5.467 | 0.95 |
| RF-24 | 6.446 | 0.95 | 5.613 | 0.94 |
| RF-25 | 6.369 | 0.95 | 5.453 | 0.95 |
| RF-26 | 6.427 | 0.95 | 5.536 | 0.95 |
| RF-27 | 6.322 | 0.95 | 5.547 | 0.95 |
| **RF-28** | **6.318** | **0.96** | **5.565** | **0.95** |
| RF-29 | 6.375 | 0.95 | 5.480 | 0.95 |

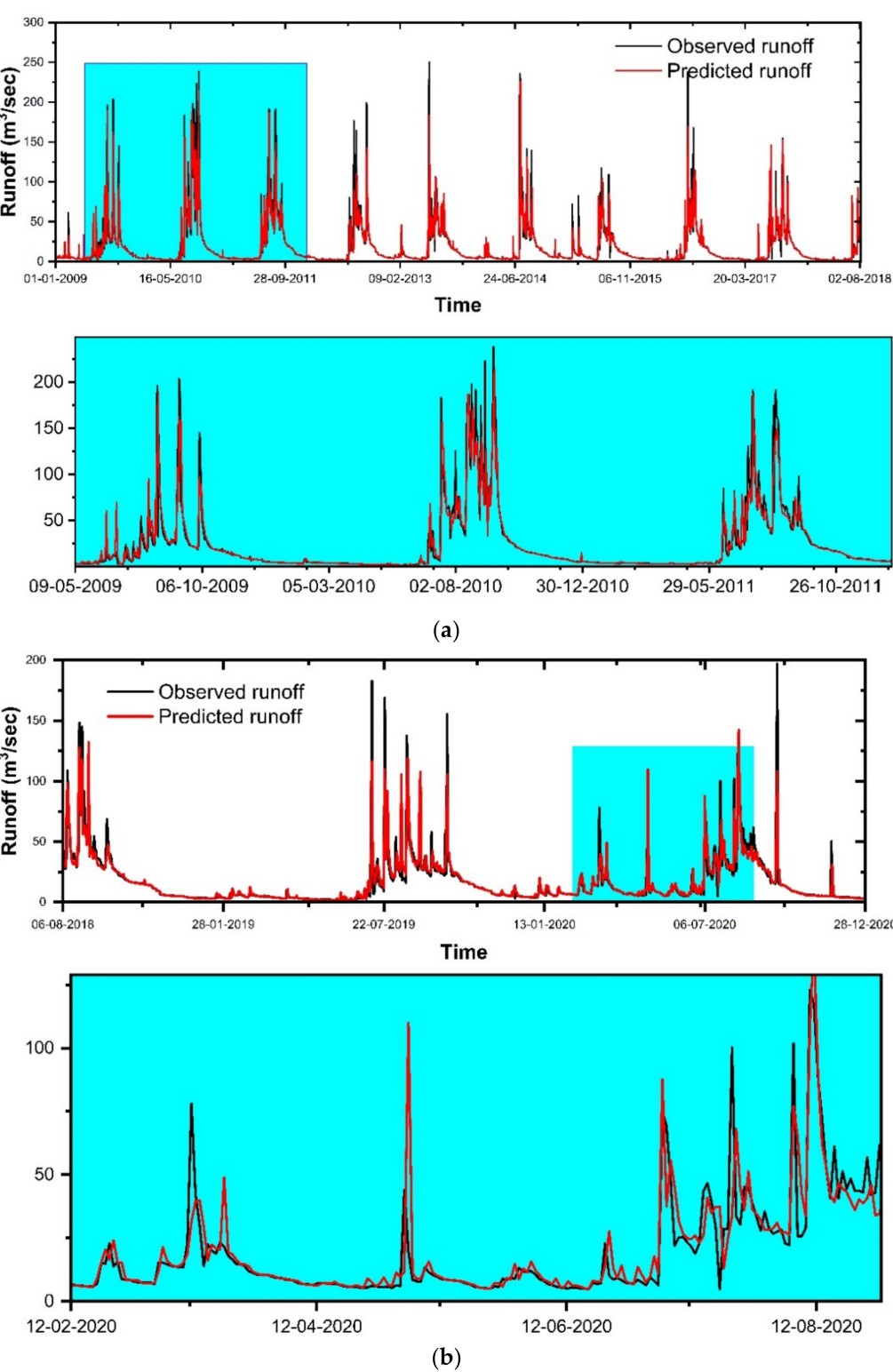

**Figure 6.** Results of simulation of the runoff (m$^3$/s) of Gola watershed using MARS model from 2009 to 2020 (**a**) training and (**b**) testing period.

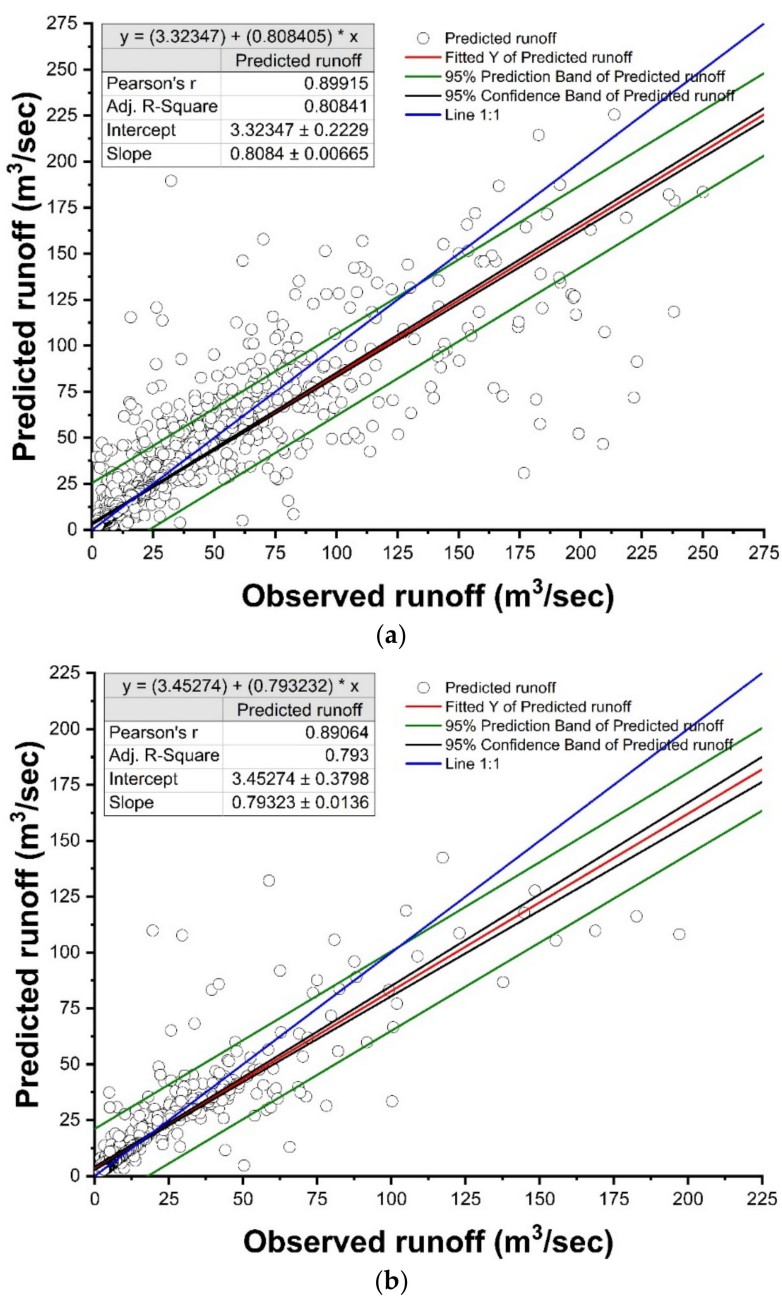

**Figure 7.** Scatter plots, evaluation statistics, and confidence intervals of the observed and predicted daily runoff Q(t) using MARS model. (**a**) Training dataset and (**b**) testing dataset.

5.2.3. SVM Model for Runoff Prediction

The present study uses the radial basis and linear kernel functions for SVM-based runoff modeling. The radial basis function performed better than the linear function, which is the reason it was selected for the current study. Because the trial-and-error method cannot achieve the best performance, optimization algorithms were used in the SVM [150].

The temporal variations and scatter plots of the observed and predicted runoff during the training and testing period are displayed in Figures 8 and 9, respectively. The time-series plot revealed the fact that the model was underpredicting large values in the training period as well as in the testing period (Figure 8a,b). The values of RMSE, $R^2$, NSE, and PBIAS for the SVM model were 12.614 m3/s, 0.83, 0.81, and −3.90% for the training period and 14.02 m$^3$/s, 0.60, 0.60, and 0.40% for the testing period, respectively (Table 3). The $R^2$ value (0.83) shows a strong linear relationship between the observed and predicted variables in the training period. It was found satisfactory (0.60) during the testing period. The NSE

value (0.81) revealed good model predictive skills during training. The 0.60 value in the testing period shows satisfactory predictive skills during the testing period. The PBIAS value was found to be −3.90% during the training period, which shows the model was overpredicting the runoff values during the training period, and the testing period (0.40%) reveals that the model was underpredicting the runoff values. According to Figure 9a,b, using 95% confidence intervals, the results showed that some of the points of the simulated runoff values (m³/s) are outside of the confidence range, which indicates underestimation in the training dataset and overestimation and underestimation of the target points in the testing dataset. However, the model's performance is acceptable according to the presented results.

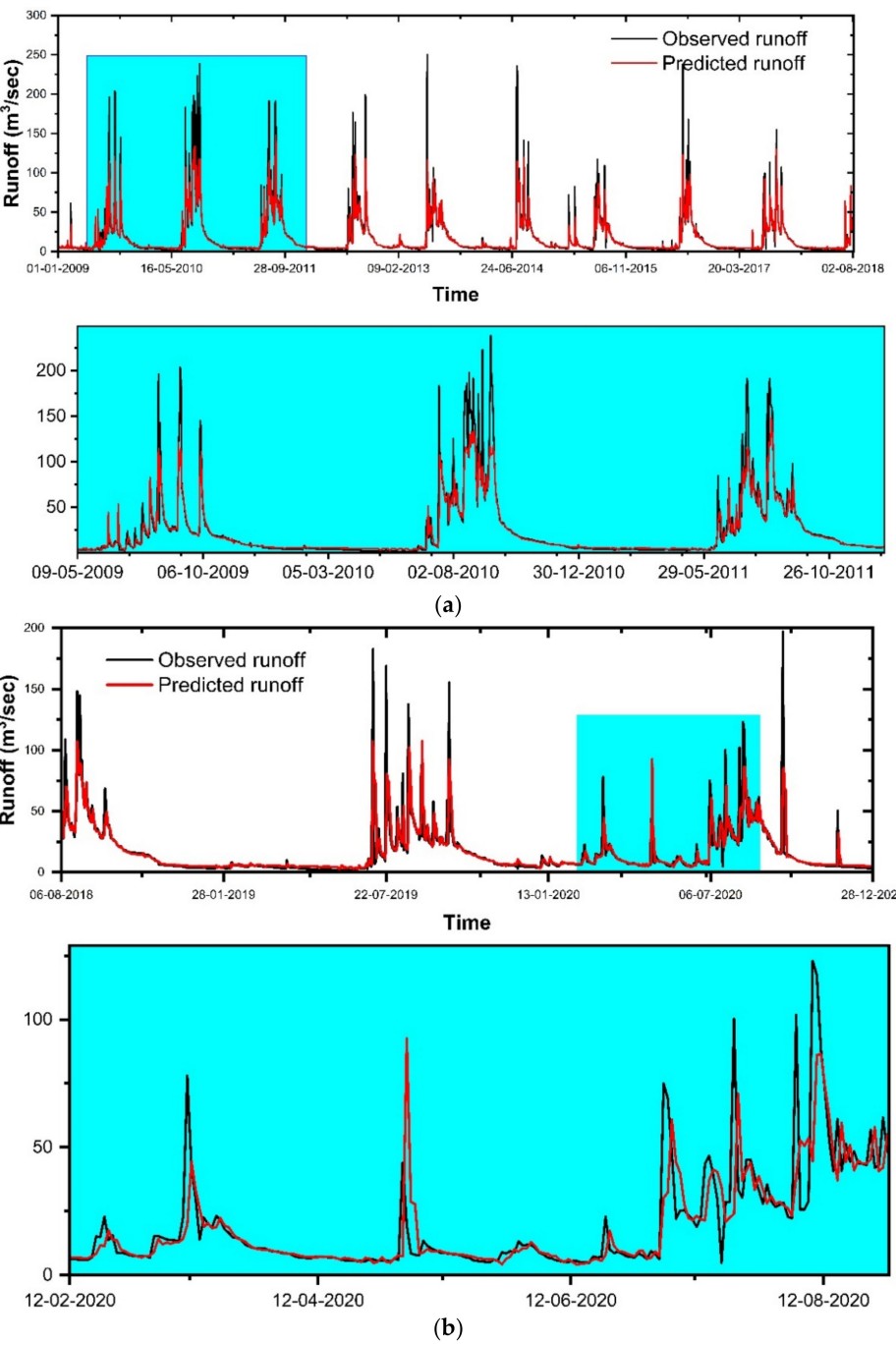

**Figure 8.** Results of simulation of the runoff (m³/s) of Gola watershed using SVM model from 2009 to 2020 (**a**) training and (**b**) testing period.

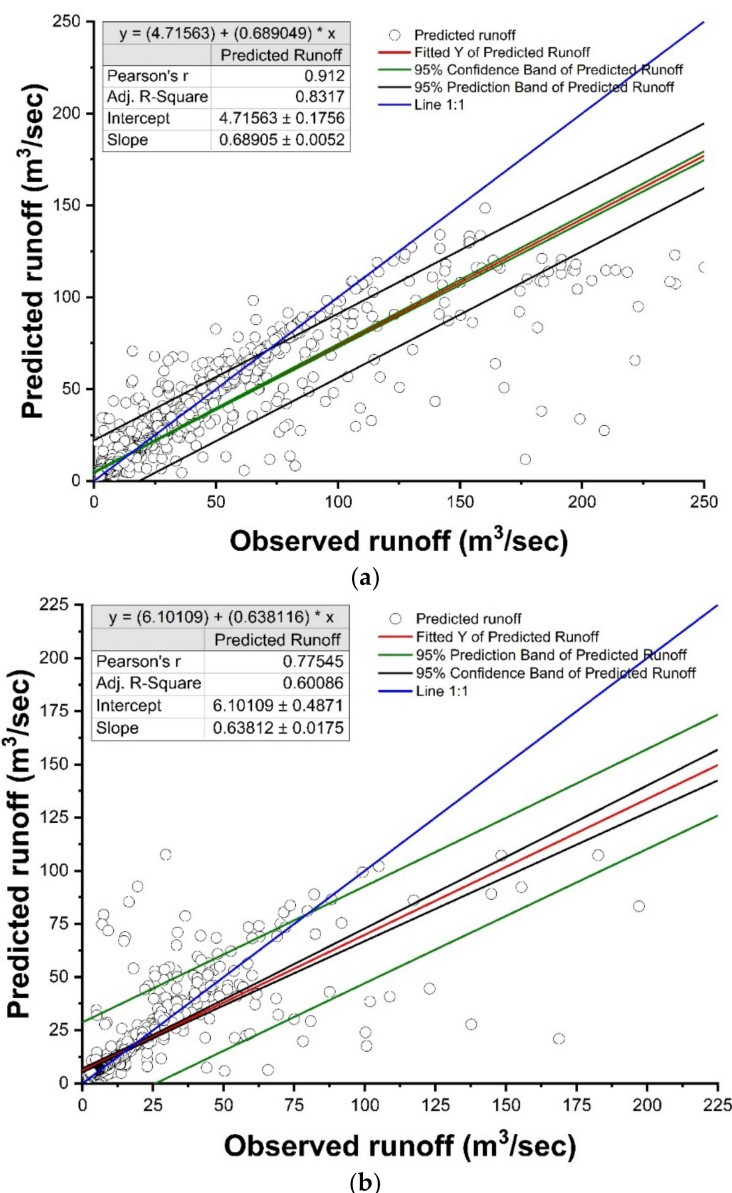

**Figure 9.** Using SVM model, scatter plots, evaluation statistics, and confidence intervals of the observed and predicted daily runoff Q(t). (**a**) Training dataset and (**b**) testing dataset.

### 5.2.4. Random Forest Model for Runoff Prediction

Two parameters were used in tuning the random forest model *ntree* (number of trees) and *mtry* (number of variables) [150]. In the present study, a trial-and-error technique was used in which the *n-tree* values varied from 200 to 600, and the *m-try* values varied from 2 to 6 to find the best performing random forest model. It was found that 400 decision trees (*n-tree*) and seven variables (*m-try*) were optimal for the best model fit. It was apparent from Table 4 that the RMSE values were in the range of 6.318 m$^3$/s to 6.480 m$^3$/s and $R^2$ values were in the range of 0.95 to 0.96 during the training period. The values of RMSE lay in the range of 5.430 m$^3$/s to 5.677 m$^3$/s, and $R^2$ values lay in the range of 0.94 to 95 during the testing period. From the evaluation of all of the results, it was observed that the RF-28 model was superior to the other RF models.

The RMSE, $R^2$, NSE, and PBIAS values of the RF-28 model were 6.318 m$^3$/s, 0.96, 0.94, and −0.20% for the training period and 5.565 m$^3$/s, 0.95, 0.92, and −0.10% for the testing period (Table 3). The low RMSE values show a concentration of data around the best fit line. The $R^2$ values (0.96 and 0.95) during the training and testing period revealed a strong linear relationship between the observed and predicted runoff values. The NSE values were

found to be 0.94 and 0.95 during the training and testing periods, respectively, which shows the good predictive ability of the model. The PBIAS values revealed that the model slightly overpredicted the runoff values during training and testing. The temporal variations and scatter plots of the observed and predicted runoff during the training and testing periods are displayed in Figures 10 and 11, respectively. The time-series plot revealed the fact that the model slightly overpredicted in the training period as well as in the testing period (Figure 10a,b). The simulation results of the RF model are also shown in Figure 11a,b: except for a few overestimated and underestimated cases in the testing period, all of the simulated data are in the 95% confidence intervals. The model's accuracy is also confirmed. According to the statistics presented in Figure 11a,b, it can be concluded that the RF model has a high ability to simulate the runoff value.

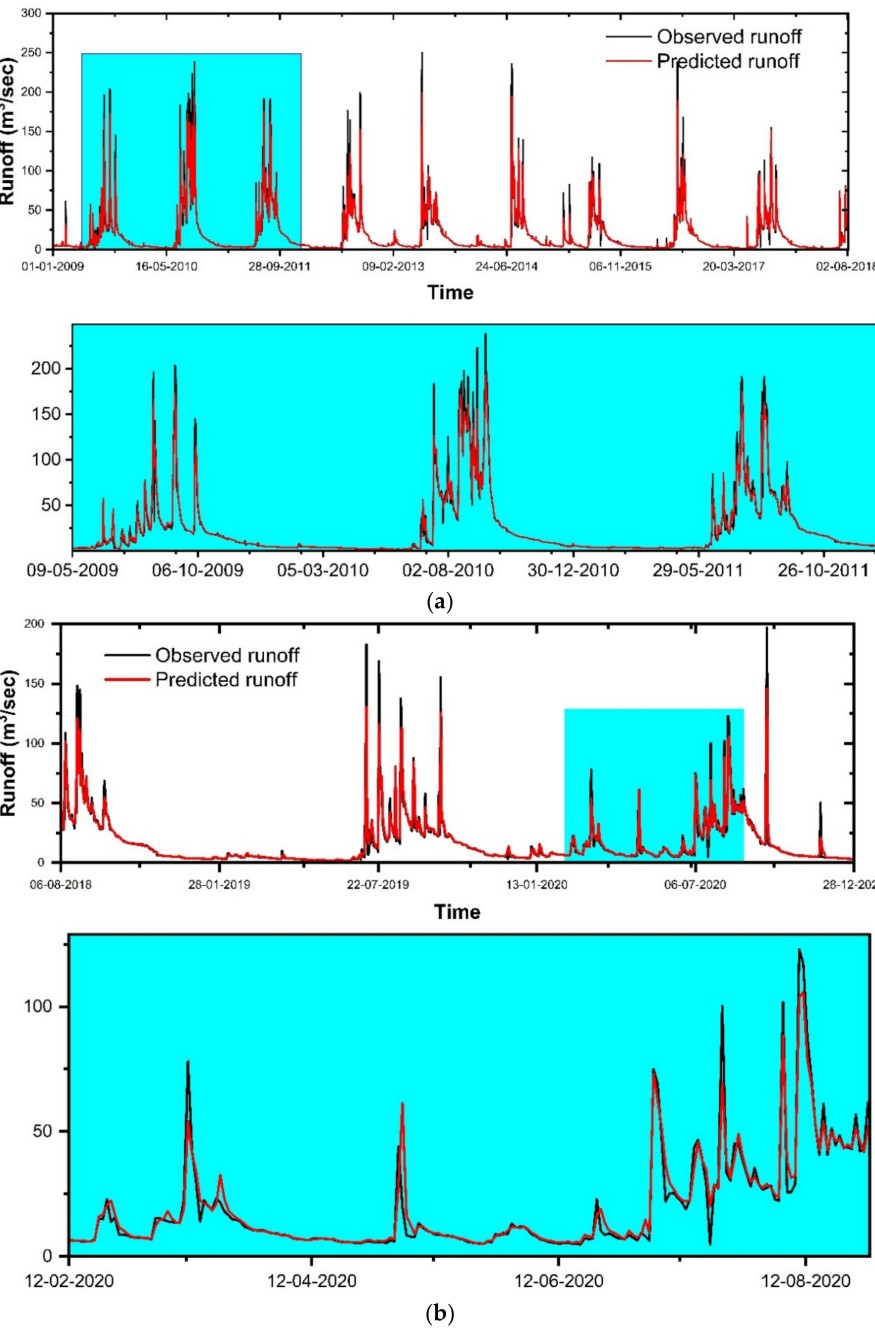

**Figure 10.** Results of simulation of the runoff (m$^3$/s) of Gola watershed using RF model from 2009 to 2020 (**a**) training and (**b**) testing period.

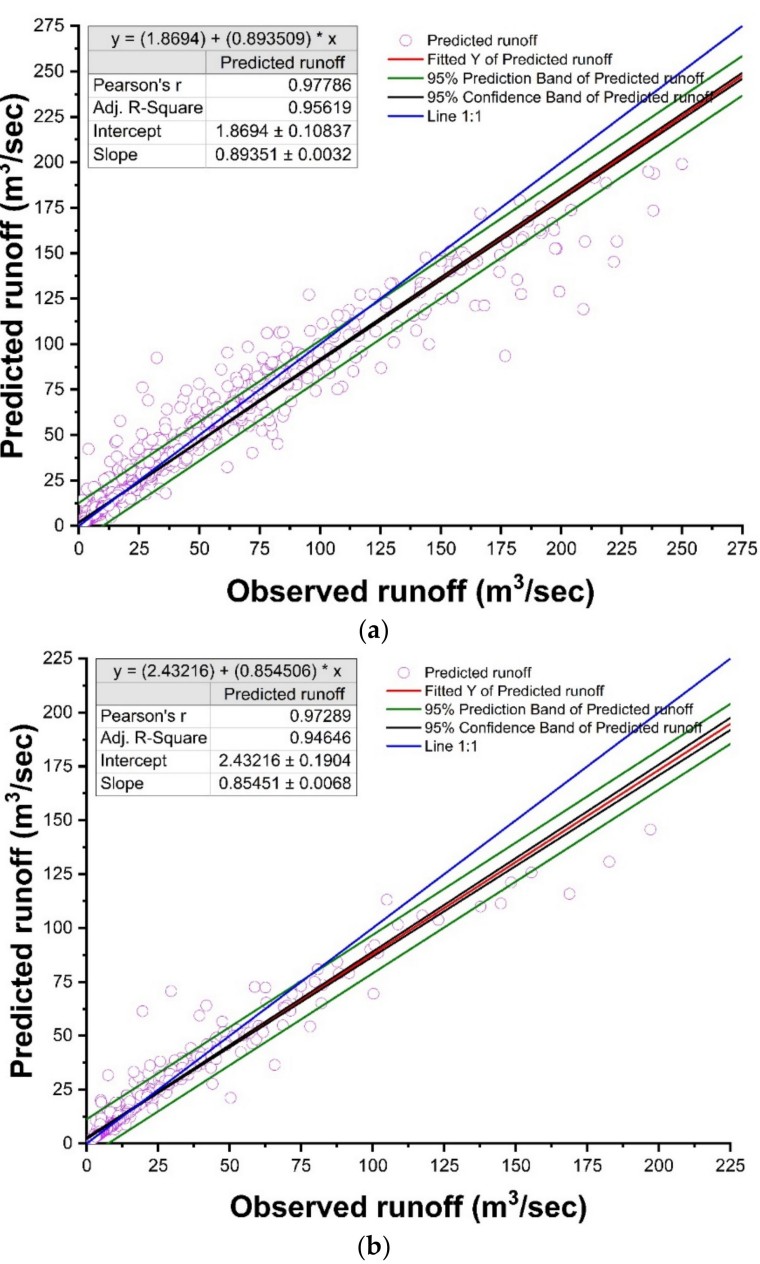

**Figure 11.** Using RF model, scatter plots, evaluation statistics, and confidence intervals of the observed and predicted daily runoff Q(t). (**a**) Training dataset and (**b**) testing dataset.

*5.3. Model Comparison*

It was noticed that the MLR model showed the least accuracy among all models (Table 3). As reported by Panda et al. [151] (2022), the poor performance of the MLR model might be attributed to the following two reasons: (1) the inability of the MLR to address predictor–predictand nonlinearity, and (2) the reduced efficiency of the model due to the presence of outliers and serial correlation. The RF-developed hydrological model exhibited significantly improved performance over the rest of the models. The superior performance of the RF model could be due to the following reasons: (1) the superior ability of the model to address nonlinearity compared to the rest of the models [152], (2) its capacity to handle noisy data efficiently [153], and (3) its ability to reduce the overfitting problem [154]. It was also found that the MARS model outperformed the SVM model during the testing period. This indicated that the MARS could handle the predictor–predictand nonlinearity better than the SVM model.

The violin plot distribution of the observed and simulated runoff during the training and testing periods is depicted in Figure 12. The MARS model captured the extreme values better during the training than the other models. However, the RF model demonstrated a greater ability to capture the high runoff values during the testing period. This indicated that the RF model could learn the hidden processes better than the other models. The performance of the MARS model was similar to that of the RF. Although the MLR performed better in capturing extreme events during the calibration period, it could not perform similarly during the validation period. The SVM model showed the least efficacy in simulating high values during the calibration and validation periods.

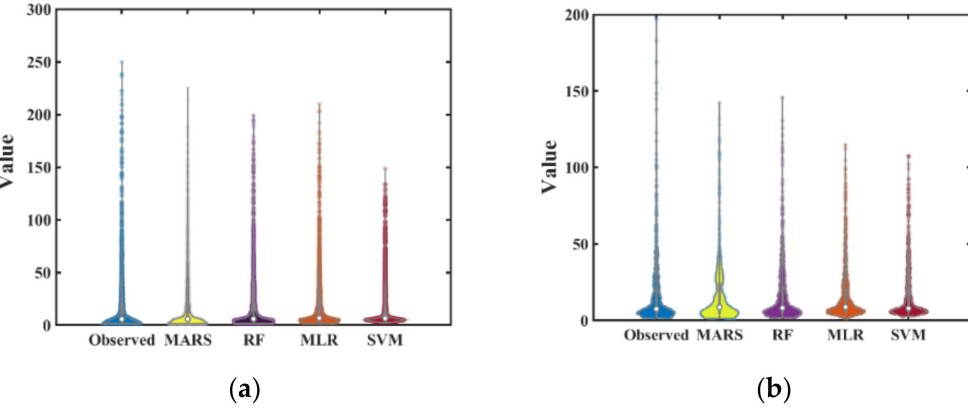

(**a**)                                    (**b**)

**Figure 12.** Violin plot displays the observed and predicted runoff distribution for the four models during the (**a**) training and (**b**) testing phase at the Gola watershed.

The relative error plot further validated the above results (Figure 13). Finally, the model efficiencies were compared using a Taylor diagram (Figure 14). It was concluded that the RF model showed the highest accuracy, followed by the MARS model. In contrast, the SVM model showed the lowest efficacy, followed by the MLR model.

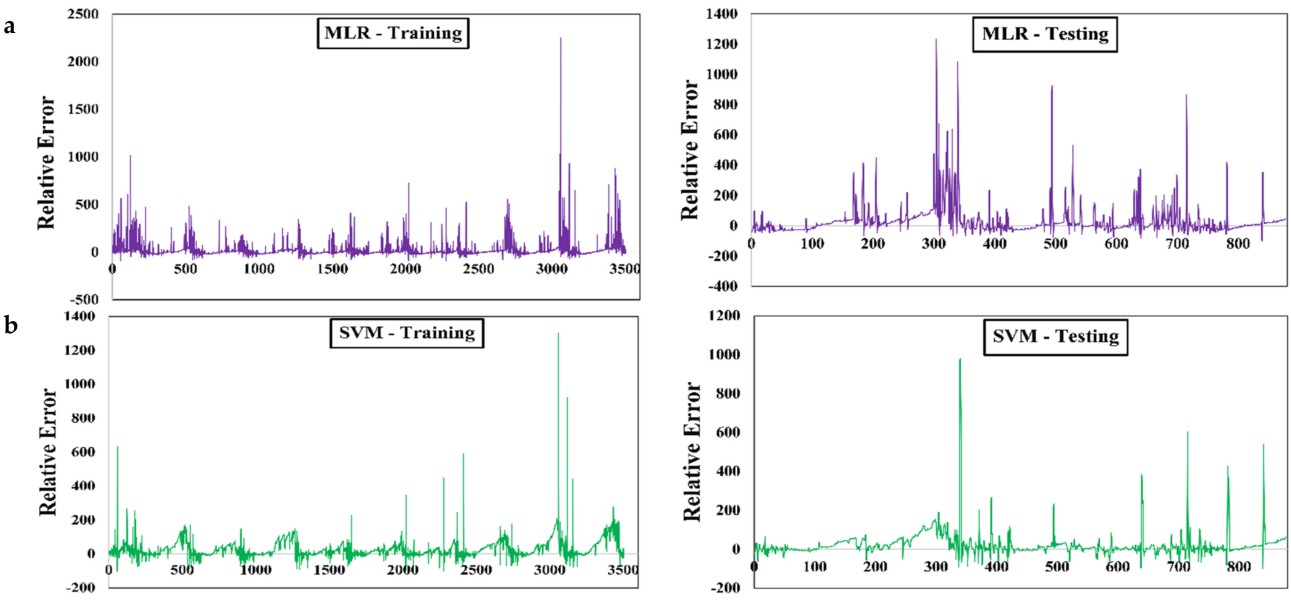

**Figure 13.** *Cont.*

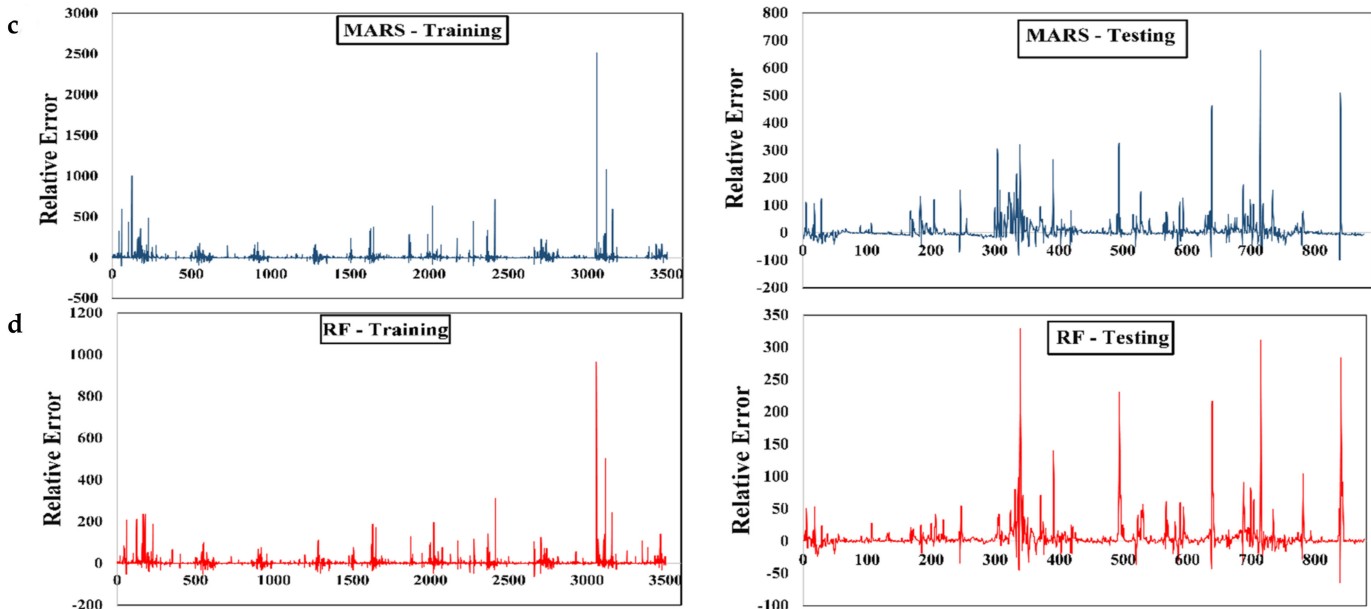

**Figure 13.** Relative error distribution over the training and testing phase for the daily timescale river flow for the Gola watershed, (**a**) MLR, (**b**) SVM and (**c**) MARS, and (**d**) random forest.

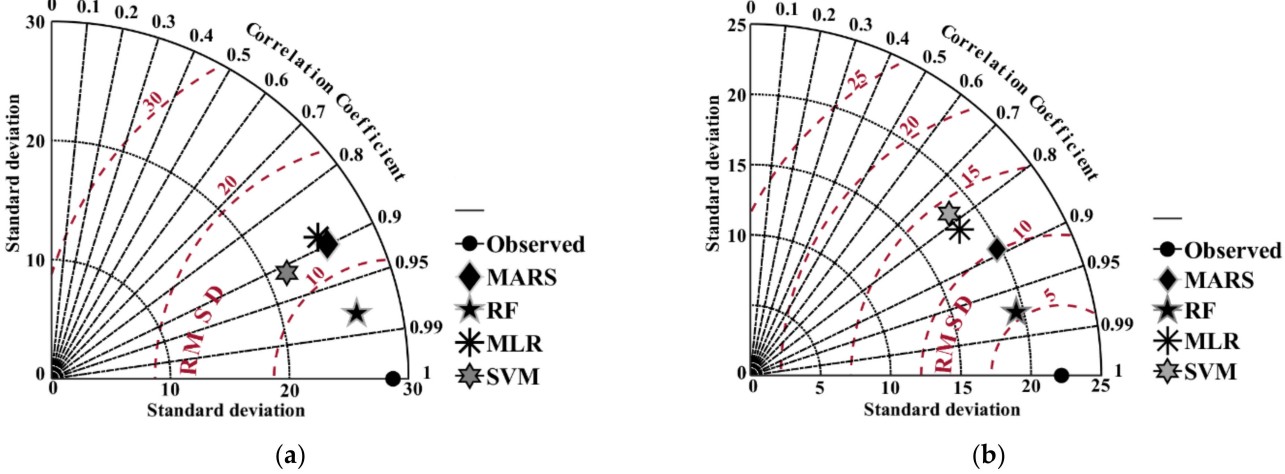

(**a**)                (**b**)

**Figure 14.** Taylor diagram of SVM, random forest, MARS, and MLR models during the (**a**) training and (**b**) testing period at the Gola watershed.

## 6. Conclusions

Rainfall is an essential hydrological phenomenon to maintain the balance of freshwater availability for the survival and growth of life. This study was conducted to evaluate the runoff pattern of the Gola watershed. The comparative results of the training and testing datasets between the MLR, MARS, SVM, and RF models' potential to predict the runoff of the Gola watershed were investigated. Among the developed models, in terms of root mean square error (RMSE), the ranking of the models was RF, MARS, SVM, and MLR for the training period and RF, MARS, MLR, and SVM for the testing period, respectively. Based on the coefficient of determination ($R^2$) statistics, the models were ranked as RF, SVM, MARS, and MLR for the training dataset and RF, MARS, MLR, and SVM for the testing dataset. The models were ranked as RF, SVM, MARS, and MLR for training and RF, MARS, SVM, and MLR for testing in the case of NSE statistics, respectively. Based on the quantitative analysis and indices, the ranking of the models was RF, MARS, SVM, and MLR for the training period, and RF, MARS, MLR, and SVM for the testing period. Perhaps these results were due to data division, input uncertainties, and model parameter

optimization. In order to determine the consistency of the models, these should be tested using varying data lengths and training–testing splits. The obtained results suggested that the accuracy of the MLR, MARS, SVM, and RF techniques were adequate using rainfall and runoff parameters for modeling. It was found that there was variation in the results of different machine learning models. The evaluation of the model performance revealed that the RF model outperformed the other regression models for predicting the runoff of the Gola watershed.

**Author Contributions:** Conceptualization, A.K.S. and P.K.; methodology, A.K.S.; software, A.K.S.; validation, P.K. and D.K.V.; formal analysis, A.K.S. and K.S.K.; investigation, P.K. and D.K.V.; resources, P.K.; data curation, A.K.S.; writing—original draft preparation, A.K.S., D.K.V., K.C.P., A.S. and K.S.K.; writing—review and editing, D.K.V., R.A., A.E., A.K. and S.H.; visualization, D.K.V. and E.M.; supervision, P.K. and D.K.V.; project administration, A.K.S.; funding acquisition, N.A.-A. All authors have read and agreed to the published version of the manuscript.

**Funding:** This research received no external funding.

**Institutional Review Board Statement:** Not applicable.

**Informed Consent Statement:** Not applicable.

**Data Availability Statement:** The authors would like to collaborate, and data can be available upon request.

**Acknowledgments:** The authors are grateful to the Department of Soil and Water Conservation Engineering, G.B. Pant University of Agriculture and Technology, Pantnagar, Uttarakhand, India, and to Gola Barrage gauge station Haldwani–Kathgodam, Uttarakhand, India, for providing data for this research. Alban Kuriqi acknowledges the Portuguese Foundation for Science and Technology (FCT) for their support through PTDC/CTA-OHR/30561/2017 (WinTherface).

**Conflicts of Interest:** The authors declare no conflict of interest.

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
