# Peer review of "An Integrated Statistical-Machine Learning Approach for Runoff Prediction"

_sustainability, doi:10.3390/su14138209_

Round 1
Reviewer 1 Report
I think the application of several method is a good exercise, but I have some doubts about the input data.
Is not clear to me if it was done some analysis about the rainfall gauge series. I work daily with data and have a 12 years of continuous data with no failure seems strange. Moreover, as the authors explained about climatic condition, the rainfall is concentrated in the monsoon period so I suspect that the are observed high rainfall rate and consequently possible gauging errors should be present. Could this aspect influence the conclusion?
Secondly, you used Thiessen method to evaluate the mean area rainfall. Do you think that use another method can influence the results?
Finally, I'm not an expert in AI or ML but I suppose that the major hypothesis is that the watershed does not change. In the sense that is the general condition of morphology (human influce) or precipitation's characteristc (climate change) are steady for all 12 years long?
Reviewer 2 Report
Dear Authors
I read your manuscript on the comparison of a several data driven approaches with great interest. Unfortunately, I came across a couple of issues, which should in my opinion be resolved before the manuscript can be published.
The most important concern about the manuscript is its innovation. All models used in this manuscript have been used separately or comparatively in different researches. This study only examined the number of lags in model input. You need to talk about the innovation and the difference between your research and other researchers. There does not seem to be much innovation in research.
There are some other cases as follows.
In the abstract, refer to the statistical period and the data used.
Authors need to add one statement on novelty of the study and how the outcomes of the study contribute to scientific and societal advancement.
More importantly, how geographically generalizable are your findings? Have you tried testing your work in, say, other regions sharing the same climate zonation as your study region?
How confident that your conclusion about the superiority of the RF approach is supported if more lags are considered. In general, for all the studied models, can further lag change the modeling results?
It was better to estimate the parameters of the SVM model using an optimization algorithm. Because the trial-and-error method can not achieve the best performance. Since the purpose of the study is to compare different methods, the models should be developed in the best way to be comparable. Such as the following article in SVR model calculation:
Nazeri Tahroudi, M., Mirabbasi, R., Ramezani, Y., & Ahmadi, F. (2022). Probabilistic Assessment of Monthly River Discharge using Copula and OSVR Approaches. Water Resources Management, 36(6), 2027-2043.
Reviewer 3 Report
This is an interesting paper to read where the authors attempted to develop rainfall-runoff relationship to predict the runoff time series using various statistical methods. However, I have few comments on the manuscript and clarifications from the authors.
1. For the introduction sections, there were lack elaborations on the cited statements which cause the connection among the sentences could not be defined. Despite of including numerous number of citations references, it is suggested to only include highly relevant references and elaborate the citations accordingly to clarify the arguments.
2. Some sentences are hanging and incomplete. Please recheck.
3. Different name/spelling of the watershed used. In the texts, "Gola" was used but in Figure 1, it is "Gaula". Please make sure consistent name is used.
4. Lack of focus on why the study is important and has to be done to the Gola watershed? What is the main problem to be solved.
5. Research gaps related to objectives were not mentioned which is important.
6. It is recommended to include the locations of the rainfall stations, streamflow station, and the river outlet in the map in Figure 1.
7. Section 2.3, entire section 3, and entire section 4 are not the methods developed by the authors. They were the existing statistical methods. Hence, these sections should be the background theories of R-studio (the software application used in the study). Suggested to add additional section of background study/theories for these section.
8. Authors did not explain based on what basis the predictors are selected. This is important because the authors only make used of rainfall and flow data while there are other factors influencing the rainfall-runoff relationship such as terrain slopes, surface runoff coefficient (landuse is the major effect), infiltration, and others. This is crucial as the explanation is relevant to the reliability of the predictive measures.
9. It is known that observed/gauged hydrological data often contains quite numbers of missing data. This definitely affect the reliability of the analysis output. However, authors did not mention about this restriction. Does it means there were no missing data throughout the 12 years. If there are, how the authors deal with it.
8. Since authors were using R-studio software application to perform the study, they are required to include explanation on how the analyses were conducted using R-studio.
9. First paragraph in section 5 were more like methods and literatures not the results.
10. In section 5.2.1 second paragraph, the choice of words "larger values" and "lower values" may be replaced by "higher flow/runoff" and "lower flow/runoff".
11. R2 and regression are not suitable for scatter plot comparison because for model performance analysis, it is not about getting the equation anymore but to see how the data falls in reference to the 1:1 line. Should use confidence interval of 95% and the lines for acceptable range in reference to 1:1 line. For example 1:2 for 20% acceptable range in the data pool.
12. Why in VSM the training set was presented for the runoff series not he testing sets like the other methods output?
13. It is recommended to use the same line types of observed and predicted plots to avoid confusion.
14. Since the study is about developing/establishing a relationship/equation in predicting runoff/flow series, the term rainfall-runoff modelling might not be appropriate as modelling requires calibration and validation processes involving various hydrological, topographical and basin characteristics which are not performed in this study. Suggest to use the term development of predictive rainfall-runoff relationship/approach using statistical methods with machine learning. Title has to be improved.
In overall, I find this research is sufficient for publication with the above corrections done. I hope my comments are able to assist the authors to improve the manuscript. Thank you.
Round 2
Reviewer 2 Report
Dear Editor;
The authors have responded well to the concerns of the reviewers, and the current version of the manuscript is upgraded from the previous version. Under these conditions, the manuscript can be published with the present form.